# PromptST: Abstract Prompt Learning for End-to-End Speech Translation

**Tengfei Yu**[1][*], **Liang Ding**[2], **Xuebo Liu**[1][†], **Kehai Chen**[1],
**Meishan Zhang**[1], **Dacheng Tao**[3], **Min Zhang**[1]

[1]Institute of Computing and Intelligence, Harbin Institute of Technology, Shenzhen, China
[2]JD Explore Academy, JD.com, Beijing, China
[3]The University of Sydney, Sydney, Australia

22s151166@stu.hit.edu.cn, liangding.liam@gmail.com, dacheng.tao@gmail.com,
{liuxuebo, chenkehai, zhangmeishan, zhangmin2021}@hit.edu.cn

## Abstract

An end-to-end speech-to-text (S2T) translation model is usually initialized from a pre-trained speech recognition encoder and a pre-trained text-to-text (T2T) translation decoder. Although this straightforward setting has been shown empirically successful, there do not exist clear answers to the research questions: 1) how are speech and text modalities fused in S2T model and 2) how to better fuse the two modalities? In this paper, we take the first step toward understanding the fusion of speech and text features in S2T model. We first design and release a 10GB linguistic probing benchmark, namely *Speech-Senteval*, to investigate the acoustic and linguistic behaviors of S2T models. Preliminary analysis reveals that the uppermost encoder layers of the S2T model can not learn linguistic knowledge efficiently, which is crucial for accurate translation. Based on the finding, we further propose a simple plug-in prompt-learning strategy on the uppermost encoder layers to broaden the abstract representation power of the encoder of S2T models. We call such a prompt-enhanced S2T model *PromptST*. Experimental results on four widely-used S2T datasets show that PromptST can deliver significant improvements over a strong baseline by capturing richer linguistic knowledge. Benchmarks, code, and scripts are freely available at https://github.com/ytf-philp/PromptST.

## 1 Introduction

Different from a cascade of separately trained automatic speech recognition (ASR, Yu and Deng 2016) and machine translation (MT, Luong et al. 2016) models, end-to-end speech-to-text translation (S2T, Duong et al. 2016; Bérard et al. 2016) directly translates source-language acoustic speech signals into a foreign text without any intermediate output, which has gained increasing popularity and obtained great success recently (Anastasopoulos and Chiang, 2018; Ansari et al., 2020; Li et al., 2021b; Bentivogli et al., 2021). Since directly modeling speech-to-text mapping is nontrivial, the common practice (Wang et al., 2020c, 2021) trains a well-performed S2T system by initializing the encoder and decoder with pre-trained single-modality models that are designed for ASR (e.g., wav2vec (Baevski et al., 2020)) and MT (e.g., mBART (Liu et al., 2020b)), respectively.

While this straightforward setting has been shown empirically successful, there are no clear answers to the research questions: 1) how does the S2T model combine speech and text modalities? 2) how to improve the fusion of speech and text modalities in the S2T model? Several attempts have been made to alleviate cross-modal representation discrepancy. For example, Yin et al. (2023) use implicit guidance from external ASR model, Le et al. (2023) optimize the CTC loss at the pre-training stage, Ye et al. (2022) use cross-modal contrastive learning (Rao et al., 2023a). Additionally, Fang et al. (2022) mix up the speech and text representation to fuse speech and text modalities from the neural representation perspective. However, these methods have not delved into analyzing the intrinsic properties of model representations.

Different from the modal fusion in the input, this paper takes the first step toward understanding the fusion of speech and text features in the S2T model. Specifically, we design a fine-grained linguistic probing benchmark, namely *Speech-Senteval*, for the S2T model following Conneau et al. (2018). Considering the information flow transferring from audio to text, the encoder of the S2T model probably learns to extract, align, and fuse acoustic features at the lower layers and then turns to learn the important knowledge, e.g., linguistic properties, to achieve translation at the high-level layers. To fur-

---

[*]Work was done when Tengfei was interning at JD Explore Academy.
[†]Corresponding Author

ther understand this, we investigate a fine-grained comparison of acoustic and linguistic analysis and leave the vanilla text probing benchmark for the T2T model. The result shows that the pre-trained S2T model fails to effectively learn linguistic information at the high-level layers, which brings a gap between speech and text in end-to-end S2T models.

Motivated by these findings, we argue that broadening the representation power of the high-level layers for the pre-trained S2T models is at the core of achieving better performance. Plenty of works have shown that translation models can largely benefit from the enriched representation (Wang et al., 2019; Wu et al., 2019; Wei et al., 2020; Liu et al., 2019a, 2020a, 2021e; Sun et al., 2022; Zan et al., 2022). To this end, we design a strategy, *abstract prompt*, to augment the high-level layers of the pre-trained S2T model, leveraging prompt-learning methods (Li and Liang, 2021). This strategy is named PromptST due to its succinct and plug-in properties. We experiment with the method in a widely used pre-trained S2T model (Wang et al., 2021) on CoVoST-2 data sets, spanning English-German, English-Catalan, English-Arabic, and English-Turkish language pairs. PromptST consistently and significantly outperforms a strong baseline by an average of +0.4 BLEU.

Our **main contributions** are as follows:

- We extend earlier works on text probing tasks to speech scenarios, organized by the type of linguistic properties they probe. We publicly release our probing benchmark *Speech-Senteval*, with the hope that it will be helpful for further study on linguistic properties of ASR and ST communities.

- By acoustic and linguistic probing analysis, we show there are great differences between speech-to-text translation and text-to-text translation, particularly in the higher-level encoder layers of the models.

- Based on our findings, we propose a straight-forward prompt learning strategy to enhance the representation capabilities of the higher-level layers in pre-trained S2T models.

## 2   Related Work

**Speech-to-Text Translation**   Most studies have been conducted to enhance end-to-end S2T models. Le et al. (2020) propose a multi-task learning approach that jointly performs automatic speech recognition and S2T, while Liu et al. (2019b) presents a knowledge distillation technique (Deng et al., 2023) by transferring knowledge from T2T models. However, previous works indicate that their successes heavily rely on large amounts of labeled training data, which is challenging to acquire. Recent advancements in pre-trained models, such as wav2vec2.0 (Baevski et al., 2020) and mBART (Liu et al., 2020b, 2021c,d), have enabled the utilization of large amounts of unlabeled data for pre-training, followed by fine-tuning on S2T tasks. By using pre-trained weights to initialize the S2T structure, the convergence accuracy and training performance of S2T models can be significantly improved (Stoian et al., 2020; Wang et al., 2021; Ouyang et al., 2022; Yin et al., 2023). In this study, we aim to investigate the impact of pre-trained knowledge on end-to-end S2T models.

**Interpreting the Neural Network Models**   Probing task is often designed to facilitate comparison between different models at a fine-grained level. For example, Yang et al. (2019) design a word reordering detection task to evaluate a model's ability to extract word order information. Rogers et al. (2021); Lin et al. (2019) analyze the hierarchy of linguistic information in transformer encoder's hidden representation. Shi et al. (2016) assess the learned representations in machine translation models for syntactic knowledge (Wang et al., 2023). Conneau et al. (2018) introduce a set of tasks to probe the linguistic knowledge encoded in sentence embeddings. These methods help open the black box of networks. In the realm of speech-to-text, Tang et al. (2021) compare auxiliary text translation tasks, and Xu et al. (2021) define the localness of a word to analyze the model architecture. In this paper, we extend these ideas and construct a 10GB probing benchmark, *Speech-Senteval*, to analyze the learned linguistic properties of S2T models.

**Prompt Learning**   Prompting refers to adding a snippet of natural language text to unlabeled data during the pre-training phase. In the case of discrete prompting, discrete information is added to the dataset as described in (Schick and Schütze, 2021). For example, discrete prompt tokens such as "It", "is", "[MASK]" can be used to classify a movie review. To illustrate, given the input text $\mathbf{x}$ ="Amazing movie!", the input embedding sequence would be formulated as

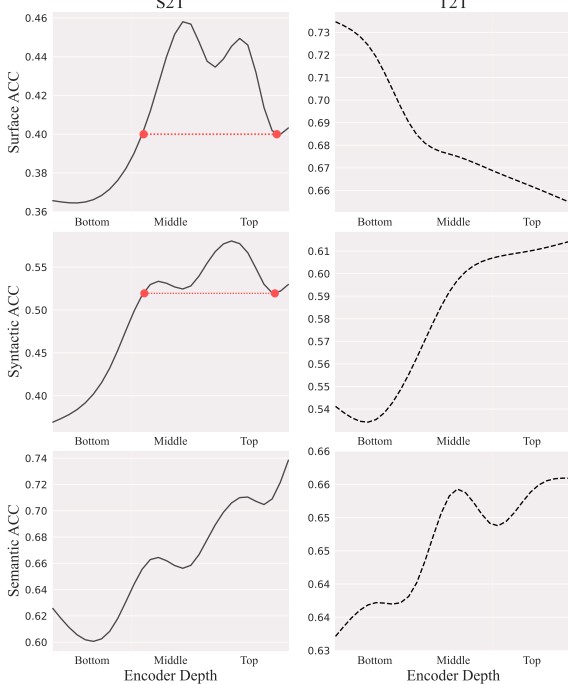

Figure 1: Performance on 10 probing tasks to evaluate the linguistics information encoded by the S2T and T2T models. These tasks are divided into three categories: *surface*, *syntactic*, and *semantic*. We averagely divide the encoder layers into three levels: *bottom*, *middle*, and *top*. The red spots show the surface and syntactic knowledge learned in the last layer is similar to that of the middle layer, indicating the S2T model can not learn linguistic information well on the top layers.

[$\mathbf{e}(\mathbf{x}), \mathbf{e}(\text{"It"}), \mathbf{e}(\text{"is"}), \mathbf{e}(\text{"[MASK]"})$]. To reduce manual intervention, Lester et al. (2021); Liu et al. (2021b); Hsu et al. (2023) introduce trainable continuous prompts (Qi et al., 2023) as a substitution for natural language prompts for language understanding tasks. For language generation tasks, Garcia and Firat (2022) uses an input template that contains a slot for input to control the output of translation models. However, the field of prompting speech translation models remains under-explored. We borrow ideas from this line of research by incorporating text-enhancement parameters into a pre-trained end-to-end S2T model to guide better speech translation.

## 3 S2T Behaviors Analysis

In a simple implementation of an end-to-end S2T model, only the top-most representation is used for decoding, ignoring the interactions among layers (Li et al., 2020). To better understand the fusion of speech and text features, we averagely divide the encoder layers into three levels: bottom, middle,

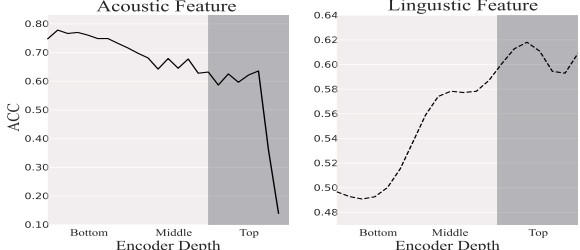

Figure 2: Performance results from acoustic and linguistic probing tasks, we align and average based on the S2T layers. The shaded areas denote our primary focus.

and top, and analyze the properties learned in each layer. Specifically, we design a fine-grained linguistic probing benchmark, namely *Speech-Senteval*, for the S2T model following Conneau et al. (2018) and leave the vanilla text probing benchmark for the T2T model to further understand linguistic properties. Subsequently, we focus on different views between acoustic (speech view) and linguistic properties (text view) in different layers of the S2T model in order to explore the acoustic and linguistic behaviors, respectively.

### 3.1 Speech-Senteval Benchmark

For conducting a comparative experiment in representation between S2T and T2T, we extend earlier work on linguistic probing tasks to S2T. In detail, we employ the same MLP classifier as Conneau et al. (2018), analyzing a rich hierarchy of linguistic properties in the encoder of S2T models[1].

**Speech-Senteval Data** To build the data set of probing tasks, we use a publicly available Baidu speech synthesis model[2] to convert the text probing data set into audio. An instance consists of a 16khz sampling audio, a transcription, and a classification label. Considering audio has more complex feature sequences than textual data, higher GPU memory constraints, and experimental costs are required, we reduce the size of the data set. The detailed data volume and statistics are shown in Appendix A.1.

**S2T and T2T Results** We analyze 10 probing tasks on S2T and T2T respectively and provide a detailed S2T statistics in Table 1. To make the analysis more intuitive, we average and smooth the experimental results according to the class the probed property belongs to. As shown in Figure 1,

---

[1]The encoder is initilized by using the wav2vec2.0-large model which is pretrained on Libri-960hr and self-trained on Libri-light (LV-6k).

[2]https://ai.baidu.com/tech/speech/tts

| Categories | Task | Emb | 4 | 8 | 12 | 16 | 20 | 24 |
|---|---|---|---|---|---|---|---|---|
| Surface | Word Content | 0.0 | 1.1 | 2.2 | 1.5 | 8.0 | 11.0 | 2.5 |
| | Sentence Length | 73.0 | 69.0 | 73.0 | 76.0 | 79.0 | 78.0 | 78.2 |
| Synatactic | Top Constituent | 33.4 | 35.5 | 57.3 | 56.9 | 70.7 | 64.5 | 50.2 |
| | Tree Depth | 25.2 | 26.9 | 28.3 | 25.0 | 33.5 | 35.7 | 41.3 |
| | Bigram Shift | 52.2 | 49.5 | 57.8 | 64.9 | 63.8 | 69.0 | 67.5 |
| Semantic | Coordination Inversion | 54.7 | 52.2 | 51.8 | 48.5 | 50.8 | 58.9 | 61.6 |
| | Object Number | 71.6 | 72.7 | 76.8 | 74.7 | 78.3 | 80.0 | 81.8 |
| | Past Present | 68.9 | 64.4 | 72.0 | 82.0 | 82.1 | 80.8 | 83.8 |
| | Subject Number | 70.0 | 59.2 | 75.2 | 83.0 | 79.3 | 82.2 | 83.9 |
| | Odd Man Out | 47.7 | 51.0 | 50.1 | 55.1 | 50.4 | 53.3 | 58.3 |

Table 1: Probing task results for S2T model. We show the analysis results of every four layers.

we compare differences between S2T and T2T with surface, syntactic, and semantic information. The results of our analysis reveal that the T2T model encodes a hierarchy of linguistic information, with surface features at the bottom, syntactic features in the middle, and semantic features at the top. The S2T system gradually encodes these three kinds of properties as the information propagates through the model's layers.

Notably, it is observed that the learning of the S2T model exhibits fluctuations. Analysis of surface and syntactic information reveals a clear decline in representation after the middle layers, which means the output of the encoder in the S2T model encodes similar information as the middle layer. Conversely, this is not present in the T2T model. This suggests that the pre-trained S2T model may not effectively learn linguistic information at higher layers, potentially due to limitations in model capacity, which brings a gap between speech and text in end-to-end S2T models.

### 3.2 Comparison: Acoustic and Linguistic

For exploring speech and text modalities of the S2T model at a fine-grained level, we employ acoustic and linguistic probing tasks to discern the differences between the two modalities.

**Setting** We analyze acoustic and linguistic properties by preparing task-oriented datasets, respectively. *Phonetic probing* can tell us how much the acoustic property S2T model catches. Adopting the methodology of Belinkov and Glass (2017), we employ the TIMIT dataset, which provides time-segmented phonemes, to extract frames from alternate encoder layers for the phoneme classification task. A classifier is then trained on these features and its performance evaluated on a test

set. The constructed training sets contain 87,295 training phonemes and 32,170 validation phonemes extracted from utterances. The possible labels are 60 phone symbols included in TIMIT (excluding the begin/end silence symbol h). *Speech-senteval probing* we proposed can tell us how much the linguistic property S2T model catches. To make comparing easy, we average all tasks by layers.

**Gap between Feature Learning** Figure 2 presents a detailed comparison of the acoustic and linguistic properties. The results indicate opposing trends, particularly at the high-level layers. The ability of the S2T model to encode linguistic properties increases, while the ability to encode acoustic properties decreases gradually. This finding confirms our hypothesis that S2T models prioritize encoding of acoustic representation at low-level layers and shift towards encoding of linguistic representation at high-level layers.

### 3.3 Discussion

Based on the above-detailed analysis of the differences between (1) S2T and T2T models in Section 3.1, and (2) acoustic and linguistic features in the S2T model in Section 3.2, it is evident that high-level layers play a crucial role in undertaking linguistic representation tasks. However, the standard, pre-trained end-to-end S2T model represents linguistic features sub-optimally at higher encoder layers compared to the T2T model. Considering the importance of augmenting linguistic features for achieving high-quality translation (Dyvik, 1992; Sennrich and Haddow, 2016; Ding and Tao, 2019), we believe that enhancing the representation power of high-level layers in S2T models could exploit more linguistic properties and thus have the potential to improve speech translation performance.

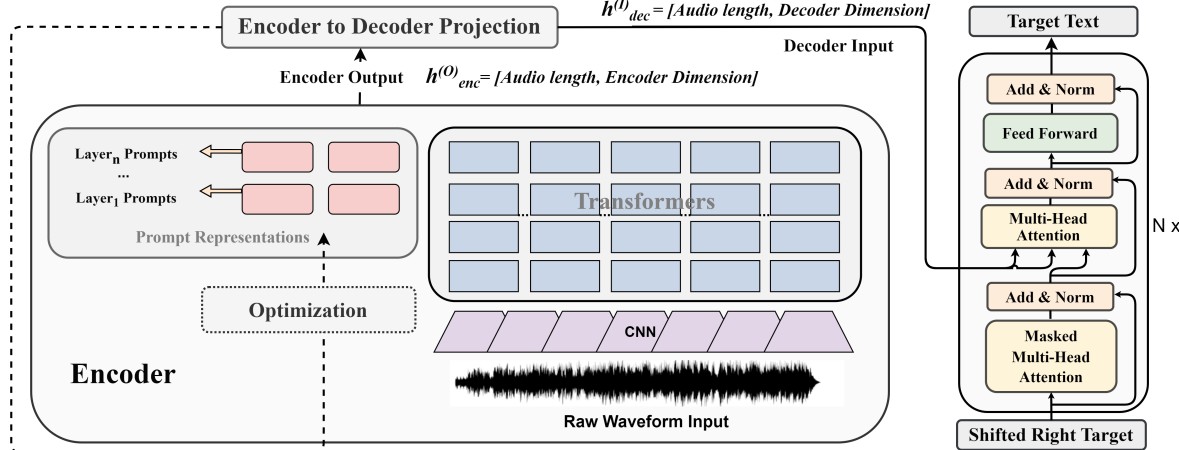

Figure 3: Architecture of the PromptST and its integration with the transformer block. Pink blocks refer to trainable prompt embeddings. For an $M$-layers encoder, we add prompt embedding from $(\frac{M}{2} + 1)^{th}$ layers. Blue blocks are transformer block embeddings initialized by wav2vec2.0. The decoder is an N-layer transformer-based model.

## 4 Abstract Prompt Learning for ST

Motivated by the analyses in Section 3, augmenting the representation power of higher encoder layers with a plug-and-play strategy is essential to achieve better linguistic knowledge that serves for high-quality speech translation. Besides efficient and plug-in properties, the method is required to enhance the higher layers without disturbing the phonetic information learned in the lower layers. Accordingly, we precisely meet these requirements by introducing a soft prompt strategy (Liu et al., 2021a) that has been empirically successful in several natural language understanding tasks but remains under-explored in the field of speech translation. Specifically, higher encoder layers in pre-trained ST models will be equipped with learnable soft-prompt representations to capture additional abstract information, e.g., linguistic knowledge. We call the proposed method *Abstract Prompt*.

### 4.1 S2T Architecture Description

Speech-to-text task directly translates source-language audio to a foreign text without any intermediate output. Intuitively, encoder-decoder (i.e., Seq2Seq) models are considered particularly suitable where the input and output sequences are not monotonically aligned.

In this study, we use pre-trained S2T architecture, consisting of a wav2vec2.0 (Baevski et al., 2020) encoder and transformer-based (Vaswani et al., 2017) decoder. Given a source raw waveform audio, we use the wav2vec2 feature processor to extract a raw waveform input, which is normalized

to zero mean and unit variance. The total stride of the encoder determines the number of time-steps $T$ which are input to the Transformer. To address the issue of dimension inconsistency between the audio and text, we integrate a two-layer multilayer perceptron (MLP) projection module between the pre-trained encoder and decoder. Mathematically, the encoder output $h_{enc}^{(O)}$ to the decoder input $h_{dec}^{(I)}$ is derived from:

$$h_{dec}^{(I)} = \text{MLP}\left(h_{enc}^{(O)}\right) \quad (1)$$

The transformed representation, $h_{dec}^{(I)}$, is integrated into the cross-attention module of the decoder.

### 4.2 Abstract PromptST

The pink blocks in Figure 3 depict the concept of Abstract PromptST, which involves the incorporation of continuous prompts in the higher layers of the S2T model encoder. The method employed in this study is not novel and can be considered an optimization of P-Tuning V2 (Liu et al., 2021a).

Technically, we formalize the pseudo prompt tokens as $[p_1, \cdots, p_m]$. PromptST maps them into trainable tensors $emb([p_1, \cdots, p_m])$. In practice, the transformer architecture generates representations of consistent dimensionality across all layers. This consistency presents a challenge when attempting to establish a direct mapping between the prompt representations and the hidden states, as the dimensions of the two may not align. To overcome this challenge, we utilize the attention module in the encoder to facilitate the integration of the prompt representations with the hidden states.

Formally, the vanilla self-attention module can be represented mathematically as

$$\text{Attn}\left(w^q \mathrm{h}_{\text{enc}}^{(i)}, w^k \mathrm{h}_{\text{enc}}^{(i)}, w^v \mathrm{h}_{\text{enc}}^{(i)}\right) \quad (2)$$

where projections $w^q, w^k, w^v$ are parameter matrices and $\mathrm{h}_{\text{enc}}^{(i)}$ is the input of the $i$-th layer.

For the selected $i$-th layer of the encoder, we concatenate the continuous prompt representations to the keys and values in the self-attention module. A randomly initialized embedding is employed to achieve our objectives:

$$\begin{aligned} \text{keys}^{(i)} &= \left[\, emb([p_1^k, \cdots, p_m^k]);\ w^k \mathrm{h}_{\text{enc}}^{(i)} \right] \\ \text{values}^{(i)} &= \left[\, emb([p_1^v, \cdots, p_m^v]);\ w^v \mathrm{h}_{\text{enc}}^{(i)} \right] \end{aligned} \quad (3)$$

where $m$ is the total length of prompt representations and $emb(\cdot)$ denotes a initialized representation. This approach allows the model to assign different weights to different utterances in the input and prompt. Additionally, we discuss the performance of PromptST by incorporating a two-linear module and a *tanh* activation to reparameterize the prompt representations in Appendix A.2.

For the low-level layers of the encoder, we adopt the vanilla self-attention module. Whereas the self-attention module in high-level layers of the encoder is revised into the following form:

$$\text{Attn}\left(w^q \mathrm{h}_{\text{enc}}^{(i)}, \text{keys}^{(i)}, \text{values}^{(i)}\right) \quad (4)$$

To enhance the efficiency of fine-tuning while considering model performance, it is essential to utilize PromptST *in conjunction with* decoder fine-tuning so that the prompt and the decoder parameters can be updated jointly. We discuss the impact of different fine-tuning strategies on model performance in the Appendix A.3.

### 4.3 Experiment Setup

**Data**   We use the CoVoST-2 ST data set[3], which is a large-scale multilingual ST corpus containing both XX-English and English-XX translation tasks. It is the largest open data set available to date from the perspective of the total volume and covered languages. Specifically, we choose four language pairs, including English (En) to German (De), Catalan (Ca), Arabic (Ar), and Turkish (Tr). Each of them contains 430 hours of annotated data.

---

[3]https://github.com/facebookresearch/covost

**Models and Settings**   Our implementation is based on the HuggingFace (Wolf et al., 2020) speech encoder-decoder models. Following the best configuration of Wang et al. (2021), we use a sequence-to-sequence model. The encoder is a wav2vec 2.0 model with several convolutional layers followed by a Transformer network. The decoder is a 7-layer Transformer network whose embedding size is 256, the number of attention heads is 4, and the FFN dimension is 2048.

During model training, we reload the pre-trained models *s2t-wav2vec2-large-en-{de/ca/ar/tr}* from HuggingFace hub[4]. We use the byte-pair encoding (BPE) (Sennrich et al., 2016) implementation from Huggingface to learn the sub-word segmentation where the vocabulary size of subword tokens $\mathcal{V}$ is set as 10K. We train our model with AdamW optimizer (Loshchilov and Hutter, 2019) with a learning rate of 5e-5. We apply the label smoothing with 0.1 and layer drop with 0.05. We set the batch size as 4, which containing 4.8M tokens. We perform the gradient accumulation trick (Ott et al., 2018) with one update per 16 batch. A masking strategy similar to wav2vec 2.0 is adopted with the mask length set as 5 and the mask probability set as 0.15.

**Optimization and Evaluation**   In order to explore the influence of the method and length of prompt for speech translation, we discuss the optimization and prefix length in Appendix A.2. We choose the prefix length by searching in the interval $[40, 400]$ and without MLP reparameterization. During model decoding, we use beam search with a beam size of 5. Models are trained for 20K updates and the best checkpoint is selected w.r.t BLEU score (Papineni et al., 2002) on the valid set. All models are fine-tuned with 8 NVIDIA A100 GPUs.

### 4.4   Main Results

We use the official baselines for most prior works (Wang et al., 2020b, 2021), which are implemented upon *fairseq* (Wang et al., 2020a) and *HuggingFace*. Table 2 demonstrates our final results on the test sets. We first evaluate performance of a strong baseline model[5] (#Model "5") we analyzed in Section 3. For a fair comparison, we adopt an identical fine-tuning strategy wherein we fix the encoder and

---

[4]https://huggingface.co/facebook
[5]To evaluate the efficacy of our proposed method, we refrain from utilizing the language model for decoding as adopted in Wang et al. (2021).

| ID | Model | #Para | En-De | En-Ca | En-Ar | En-Tr | Avg. |
|---|---|---|---|---|---|---|---|
| *Previous results on CoVoST 2* | | | | | | | |
| 1 | End-to-end ST (Wang et al., 2020b) | - | 13.6 | 20.2 | 8.7 | 8.9 | 12.9 |
| 2 | End-to-end ST (+ pre-ASR) (Wang et al., 2020b) | - | 16.3 | 21.8 | 12.1 | 10.0 | 15.1 |
| 3 | Cascade SOTA (Li et al., 2021b) | - | 19.4 | 25.0 | 14.3 | 11.7 | 17.6 |
| 4 | XMEF-JT. (Li et al., 2021b) | 1.1B | 25.8 | 30.9 | 18.0 | 17.0 | 22.9 |
| *Our results* | | | | | | | |
| 5 | Wav2vec-2.0 + self-training (LV-60k) (Wang et al., 2021) | 0.3B | 26.0 | 30.7 | 19.3 | 17.5 | 23.3 |
| 6 | Continue Train | 0.3B | 25.9 | 33.3 | 19.3 | 17.6 | 24.0 |
| 7 | PromptST | 0.3B | **26.4**[‡] | **33.7**[‡] | **19.6**[‡] | **17.9**[‡] | **24.4** |

Table 2: BLEU scores on four language pairs of CoVoST-V2 test set. "[‡]" indicates that the proposed method is significantly better than continued train results at a significance level ($p < 0.05$). We also report the Translation Error Rate (Snover et al., 2006) in Appendix A.4.

| Selected Layers | En-De | En-Ca | En-Ar | En-Tr |
|---|---|---|---|---|
| 0-24 layers | 29.9 | 36.7 | 23.5 | 20.4 |
| 20-24 layers | 29.8 | 36.8 | 23.4 | 20.6 |
| 16-24 layers | 29.9 | 36.5 | 23.7 | 20.5 |
| 12-24 layers | **30.1**[‡] | **37.4**[‡] | **23.8**[‡] | **21.0**[‡] |

Table 3: BLEU scores of adding prompt representations to different layers on the valid set. "[‡]" indicates that "12-24 layers" significantly outperform "0-24 layers" with a high degree of statistical significance ($p < 0.05$).

| Model | Layers | En-De | En-Ca | En-Ar | En-Tr |
|---|---|---|---|---|---|
| Adapter | 0-24 | 28.0 | 34.9 | 21.7 | 19.6 |
| | 12-24 | 27.9 | 34.9 | 21.6 | 19.5 |
| PromptST | 12-24 | **30.1**[‡] | **37.4**[‡] | **23.8**[‡] | **21.0**[‡] |

Table 4: BLEU scores of replacing prompt with the adapter on the valid set. "[‡]" indicates our method outperforms the adapter with a high degree of significance ($p < 0.05$).

proceed to train the model with the same number of epochs as our proposed method. The continued training "Continue Train" (#Model "6") achieves 24.0 BLEU on average, which is 0.7 points higher than the baseline, showing the current model has not yet attained full convergence.

Equipping with our prompt strategy, which adds trainable tokens from 12 to 24 layers (#Model "7"), the model outperforms the prior approach in all language directions by average +1.1 BLEU points. Notably, our PromptST also surpasses continue train setting by +0.4 BLEU points, showing the effectiveness of our approach.

### 4.5 Ablation Study on PromptST

We evaluate the impact of different components of PromptST, including (1) prompting on different layers, and (2) tuning with other efficient strategies on En-De, En-Ca, En-Ar, and En-Tr datasets.

**Impact of Prompting Layer** Table 3 shows the results of equipping our Abstract Prompt on different layers. Adding soft-prompts from $12th$ layer ("12-24 layers") consistently outperforms other variants ("0-24 layers", "20-24 layers", and "16-24 layers"), which we attribute to the advantage of abstract prompt in enhancing the critical yet under-explored high-level layers. For effectiveness and simplicity, we use the upper half encoder layers "12-24 layers" as the default strategy.

**Impact of Efficient Tuning Strategy** As aforementioned, one of the reasons to employ the prompting strategy is to enhance the higher layers without disturbing the phonetic information learned in the lower layers. One may wonder whether other efficient tuning approaches, e.g., Adapter (Houlsby et al., 2019; He et al., 2022; Rao et al., 2023b), are also suitable. To answer this doubt, we investigate the impact of replacing the prompt with the adapter and adding a length adapter (Li et al., 2021a; Le et al., 2021) between the encoder and decoder to reduce speech length. The result in Table 4 shows that adding adapters to higher layers shows comparable performance with that of adding to all layers ("0-24"). Noticeably, adopting the prompt is significantly better than that of the Adapter. The reason may exactly match our guess; the adapter is akin to the series circuit when inserted into the original encoder blocks (He et al., 2021), heavily disturbing the knowledgeable information flow from lower layers, e.g., phonetic information. In contrast, prompt tokens can be analogized to the parallel circuit to provide auxiliary capacity without affecting the original information.

| Model | Surface | | Syntactic | | | Semantic | | | | | Avg. |
|---|---|---|---|---|---|---|---|---|---|---|---|
| | SeLen | WC | TrDep | ToCo | BShif | Tense | SubNm | ObjNm | SoMo | CoIn | |
| Baseline | 78.17 | 2.47 | **41.28** | 50.17 | 67.49 | 83.77 | **83.85** | 81.77 | **58.29** | 61.55 | 60.88 |
| +PromptST | **78.86** | **4.43** | 39.54 | **54.25** | **69.57** | 83.77 | 83.33 | **82.51** | 57.12 | **61.58** | **61.50** |

Table 5: Results of probing tasks on our constructed *Speech-Senteval*. We evaluate the linguistic properties learned by the up-most layer and find that our method preserves more knowledge with improved average accuracy. We also report the results of every four layers similar to Table 1 in Appendix A.5.

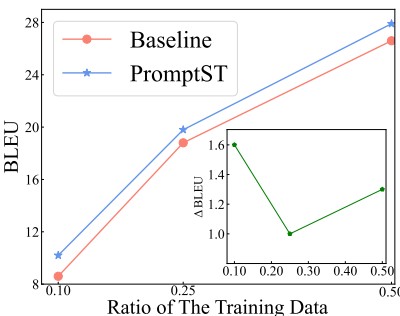

Figure 4: BLEU scores on CoVoST-2 En-Ca valid set with different ratios of the training data.

## 4.6 Analysis

We conduct a comprehensive analysis of our methodology, addressing: (1) its aptitude in learning linguistic properties, (2) its performance robustness across varying data scales, and (3) the nuanced benefits beyond the BLEU score.

**PromptST Obtains Richer Linguistic Properties** According to analyses in Section 3, our PromptST may broaden the model capacity of higher layers, affecting the linguistic properties learned by the encoder. To verify this, we select the En-De well-trained model and probe its linguistic properties. Table 5 illustrates that our method indeed preserves richer linguistic knowledge with better average accuracy (especially on the surface and syntactic, i.e., +1.4%), confirming our hypothesis.

**PromptST Robustly Works Across Data Scales** To confirm the effectiveness of our method across different data sizes, we further experiment on the En-Ca dataset partition into different data scales {0.1, 0.25, 0.5}. As seen in Figure 4, our simple method boost performances for speech translation models consistently and significantly across the different size of datasets, showing the robustness and effectiveness of our approach.

Also, researchers may doubt that our approach may fail in extremely low-resource settings *where the backbone ST model is not well pre-trained*. To dispel this concern, we conduct experiments on

| Data Set | Baseline | PromptST | △ BLEU |
|---|---|---|---|
| Ca-En | 17.85 | 18.51 | 0.66 |
| Es-En | 18.00 | 18.79 | 0.78 |
| De-En | 15.88 | 16.27 | 0.39 |

Table 6: Results of the extremely low-resource dataset. We report the results on the valid set.

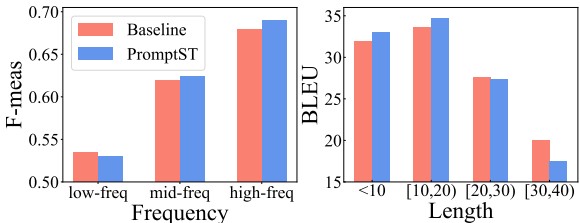

Figure 5: Word frequency and sentence length analysis.

CoVoST2.0 Ca-En, Es-En, and De-En, containing only 99 hours of annotated data on average. Table 6 shows that the baseline indeed presents an overall lower absolute BLEU score (<20) compared to the large datasets in Table 3 (∼27). However, our PromptST still significantly improves the performance by an average of 0.61 BLEU, demonstrating PromptST could be a promising plug-in strategy to provide bonuses for any data scales and correspondingly pre-trained models.

**Fine-Grained Gains Beyond BLEU Score** To understand how PromptST improves translation beyond BLEU, we use compare-mt (Neubig et al., 2019) to compare our model against baseline in terms of *word frequencies* and *sentence lengths*. Figure 5 interestingly shows PromptST (1) tends to be more robust to mid-frequency (10-1000) and high-frequency words (beyond 1000), while the baseline model performs slightly better on rare words (less than 10) and (2) facilitates shorter sentences, e.g., <20, compared to the baseline. These phenomena show PromptST can better meet the needs of daily oral communication - relatively short sentences with high-frequent words.

# 5 Conclusion

In this paper, we take the first step toward understanding the fusion of speech and text features in the S2T model by probing tasks, taking the text-to-text model as a reference. Specifically, we design and release a 10GB linguistic probing benchmark, *Speech-Senteval*, for the S2T task. We find that the uppermost encoder layers of the S2T model can not learn linguistic knowledge efficiently, which is vital for translation. Based on these insights, we propose a straightforward plug-in prompt-learning strategy, coined as Abstract Prompt, on the high-level layers to broaden the representation ability of the pre-trained ST models. Experimental results on four widely-used datasets show that PromptST can deliver significant improvements over a strong baseline by capturing richer linguistic knowledge.

## Limitations

While the proposed PromptST model augments the representation power of higher layers in the encoder, it still has some limitations: (1) our analysis primarily emphasizes the linguistic discrepancies in end-to-end S2T models, sidelining the acoustic perspective; (2) the resource-intensive nature of loading pre-trained weights from existing methods; (3) in Section 4.6, our comparison with Adapter exclusively employs the series adapter technique, neglecting the potential advantages of the parallel adapter structure as highlighted in Gállego et al. (2021), which warrants further exploration.

## Ethics Statement

In conducting our research, we strictly adhere to all applicable laws and regulations. Additionally, we take ethical considerations seriously and adhere to the standards of the ACL Ethics Policy. This paper focuses on improving speech translation. To support reproducibility, both the datasets and code used in this study are available to other researchers upon request. We have made every effort to present our findings and conclusions in an objective and accurate manner.

## Acknowledgments

This work was supported in part by the National Natural Science Foundation of China (Grant No. 62206076), Shenzhen College Stability Support Plan (Grant Nos. GXWD20220811173340003, GXWD20220817123150002), Shenzhen Science and Technology Program (Grant No. RCBS20221008093121053). Xuebo Liu was sponsored by CCF-Tencent Rhino-Bird Open Research Fund. We would like to thank the anonymous reviewers and meta-reviewer for their insightful suggestions.

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

| Tasks | Train | Valid | Test | Cate. |
|---|---|---|---|---|
| Word Content | 5,000 | 300 | 200 | 997 |
| Sentence Length | 5,000 | 300 | 200 | 5 |
| Top Constituents | 5,000 | 300 | 200 | 20 |
| Tree Depth | 5,000 | 300 | 200 | 7 |
| Bigram Shift | 7,000 | 300 | 200 | 2 |
| Coordination | 5,000 | 300 | 200 | 2 |
| Object Number | 5,000 | 300 | 200 | 2 |
| Past Present | 5,000 | 300 | 200 | 2 |
| Subject Number | 5,000 | 300 | 200 | 2 |
| Odd Man Out | 5,000 | 300 | 200 | 2 |

Table 7: Statistics of Speech-Senteval benchmark.

| Model | En-De | En-Ca | En-Ar | En-Tr |
|---|---|---|---|---|
| MLP Encoder | 29.8 | 36.9 | 23.4 | 20.7 |
| prefix-40 | **30.1** | 37.2 | 23.6 | 20.8 |
| prefix-100 | 30.1 | 37.2 | 23.6 | 20.8 |
| prefix-200 | 30.0 | 37.2 | 23.7 | **21.0** |
| prefix-300 | 30.0 | **37.4** | **23.8** | 20.8 |
| prefix-400 | 30.0 | 37.3 | 23.8 | 20.8 |

Table 8: BLEU scores on the prompt length and reparametrization (prefix-40) on valid set.

# A Appendix

## A.1 Speech-Senteval Data Set

For each task, we construct training sets containing training, test and validation data sets.[6] The overall volume of data is presented in Table 7. All sets are balanced, having an equal number of instances in each target class.

## A.2 Experiment Optimization

Previous studies have commonly employed a reparameterization encoder, such as a multi-layer perceptron (MLP), to optimize soft-prompt representations for NLU tasks. As demonstrated by Liu et al. (2021b), the use of an MLP reparameterization can enhance the robustness and performance of models. In order to evaluate the performance of the S2T model, we employ a simple MLP layer to encode the trainable prefix tokens. However, our experiments reveal that the use of an MLP led to negative effects for nearly all language pairs.

Furthermore, the prompt length has been identified as a crucial factor in the S2T model. We conducted an extensive search of the optimal prefix length for four language pairs within the range of [40, 400], respectively. As depicted in Table 8, we

---

[6]Following https://github.com/facebookresearch/SentEval/tree/main/data/probing, the dataset is made available under the BSD 3-Clause License.

| Model | En-De |
|---|---|
| Random Initialized | 1.7 |
| Finetune Encoder and Prompt | 23.6 |
| Finetune Prompt | 25.9 |
| Finetune Encoder Decoder and Prompt | 26.0 |
| Finetune Decoder and Prompt | **26.4** |

Table 9: BLEU scores for randomly initialized model and different fine-tune strategies on test set.

found that shorter prompts are sufficient for language pairs that are closely related (e.g., En-De). However, for long-distance language pairs (e.g., En-Ca, En-Ar, En-Tr), prompts longer than 200 tokens are found to be beneficial.

## A.3 Pre-training and Efficient Fine-tuning

To demonstrate the importance of pre-trained models in an end-to-end S2T model, we conduct experiments using the same settings to train a randomly initialized En-De model. As shown in Table 9, the use of random initialization results in a low BLEU score, indicating that the performance of the S2T model is heavily dependent on being initialized with pre-trained parameters.

We further evaluate the efficiency of various tuning strategies, namely: (1) fine-tuning the prompt only, (2) comprehensive fine-tuning encompassing prompt, encoder, and decoder, (3) fine-tuning the encoder with the prompt, and (4) fine-tuning the decoder with the prompt. Table 9 illustrates that fine-tuning the decoder with the prompt achieves the most optimal results. Specifically, jointly fine-tuning the encoder and prompt led to swift overfitting. This suggests that exhaustive encoder adjustments might not be ideal.

## A.4 Comparision on Translation Error Rate

In this study, we evaluate the performance of both the baseline and the PrompST models using the Translation Error Rate (TER) metric. The results, presented in Table 11, demonstrate that when our prompt strategy is applied, the model achieves an average TER score of 57.2 on the validation sets. This represents a significant reduction in the translation error rate across all language directions, with an average decline of −3.5 points, showing the efficacy of our proposed approach.

## A.5 Speech-Senteval Results on PromptST

Table 10 presents the performance of the S2T model using the PromptST approach on all 10 prob-

| Categories | Task | Emb | 4 | 8 | 12 | 16 | 20 | 24 |
|---|---|---|---|---|---|---|---|---|
| Surface | Word Content | 0.4 | 1.1 | 2.5 | 14.4 | 11.8 | 14.2 | 4.4 |
| | Sentence Length | 76.5 | 68.1 | 76.5 | 77.3 | 79.5 | 80.5 | 78.9 |
| Synatactic | Top Constituent | 32.4 | 39.5 | 48.4 | 62.2 | 67.6 | 61.9 | 54.3 |
| | Tree Depth | 19.7 | 28.5 | 28.0 | 36.0 | 31.7 | 31.2 | 39.5 |
| | Bigram Shift | 45.5 | 49.7 | 57.2 | 58.7 | 71.2 | 67.3 | 69.6 |
| Semantic | Coordination Inversion | 54.7 | 46.3 | 51.2 | 61.5 | 59.8 | 57.3 | 61.6 |
| | Object Number | 65.2 | 78.9 | 80.4 | 78.1 | 81.2 | 78.1 | 82.5 |
| | Past Present | 67.2 | 69.0 | 78.0 | 81.6 | 80.8 | 83.8 | 83.8 |
| | Subject Number | 53.7 | 65.6 | 70.4 | 79.9 | 78.8 | 81.8 | 83.3 |
| | Odd Man Out | 55.6 | 50.9 | 47.1 | 49.8 | 52.9 | 53.9 | 57.1 |

Table 10: Probing task results for PromptST. We show the analysis results of every four layers.

| Model | En-De | En-Ca | En-Ar | En-Tr | Avg. |
|---|---|---|---|---|---|
| Baseline | 56.8 | 52.4 | 67.9 | 65.5 | 60.7 |
| PromptST | **55.2** | **48.1** | **62.9** | **62.4** | **57.2** |

Table 11: Results of Translation Error Rate on valid set.

ing tasks. We run the analysis every four layers. Our experimental data clearly indicate that using PromptST enhances the model's ability to retain complex linguistic information. We specifically observe improvements in capturing surface and syntactic properties at layers 12, 16, and 24, with increases of +3.2, +1.9, and +1.4, respectively. These findings suggest that PromptST effectively expands the learning capacity of the model's higher layers, positively influencing the types of linguistic properties that the encoder can learn.