# OpenReview forum: "PromptST: Abstract Prompt Learning for End-to-End Speech Translation"
_EMNLP/2023/Conference — EMNLP 2023 Main_

### Official Review · Reviewer_68rb · 2023-08-04

**Soundness:** 4

**Excitement:**

4: Strong: This paper deepens the understanding of some phenomenon or lowers the barriers to an existing research direction.

**Paper Topic And Main Contributions:**

This paper proposes an extended approach to probing of ML model information for speech-related tasks (together with a public benchmark). Through the analysis, the authors provide a methodology to understand which information is processed where in the model (the balance between linguistic and acoustic information mostly). As a conclusion, then we see a proposed method (Abstract Prompting) that is demonstrated to enhance the representation capabilities of speech translation models (tested on multiple languages with a positive impact in BLEU scores).

**Reasons To Accept:**

The paper carries out a thorough analysis of information and probing details within layers/models that can help others in the community understand their models (and contributions) and the contribution of different input signals better. Overall validation of the technology is robust and convincing.

**Reasons To Reject:**

As identified by the authors, to me it is a shortcoming that the analysis is carried out purely from the linguistic performance point of view. Considering this is a conference focused on language tasks this is not a deal breaker, but a more balanced analysis would greatly reinforce the potential of impacting other fields.

**Reproducibility:**

4: Could mostly reproduce the results, but there may be some variation because of sample variance or minor variations in their interpretation of the protocol or method.

**Reviewer Confidence:**

3: Pretty sure, but there's a chance I missed something. Although I have a good feel for this area in general, I did not carefully check the paper's details, e.g., the math, experimental design, or novelty.

---

> ### Author Rebuttal · Authors · 2023-08-29
>
> Thank you for the review.
>
> > Q5.1 *As identified by the authors, to me it is a shortcoming that the analysis is carried out purely from the linguistic performance point of view. Considering this is a conference focused on language tasks this is not a deal breaker, but a more balanced analysis would greatly reinforce the potential of impacting other fields.*
>
> Thank you for your positive comments on our paper, and for bringing to our attention the need for a more balanced analysis of the acoustic dimensions in our study. In subsequent research, we plan to explore both acoustic aspects and machine learning in greater depth to offer a more complete analysis of the subject matter.

---

### Official Review · Reviewer_LoxQ · 2023-08-11

**Soundness:** 4

**Excitement:**

3: Ambivalent: It has merits (e.g., it reports state-of-the-art results, the idea is nice), but there are key weaknesses (e.g., it describes incremental work), and it can significantly benefit from another round of revision. However, I won't object to accepting it if my co-reviewers champion it.

**Paper Topic And Main Contributions:**

This paper is about end-to-end speech translation. The author focuses on the following two problems in E2E-ST: 1) how are speech and text modalities fused in S2T model? and 2) how to better fuse the two modalities? Preliminary analysis reveals that the uppermost encoder layers of the S2T model can not learn linguistic knowledge efficiently, which is crucial for accurate translation. Based on the finding, authors propose a prompt-enhanced S2T model, PromptST. Experimental results on four widely-used S2T datasets show that PromptST can deliver significant improvements over a strong baseline by capturing richer linguistic knowledge.

**Questions For The Authors:**

Since the parallel adapter also shows good results, I wonder how the author proposed approach differs in this case. And based on the references in the paper, it seems that the authors do not use length adapter (1D convolution) to reduce speech length. This comparison may seem unfair, since the length reduction is usually required when the adapter is applied in E2E-ST

**Reasons To Accept:**

1. Motivation is detailed and clear.
2. The s2t behaviors is well analyzed. Based on the above analysis, the authors find that high-level layers play a crucial role in undertaking linguistic representation tasks. However, the standard, pre-trained end-to-end S2T model represents linguistic features sub-optimally at higher encoder layers compared to the T2T model.
3. PromptST exceeds previous models and significantly improves model performance.
4. The experimental analysis part is  comprehensive, which is helpful to understand the mechanism of the proposed method.
5. Authors publicly release their probing benchmark Speech-Senteval.

**Reasons To Reject:**

1. There have been previous analyses of the network representation problem [1,2,3]. I also admit that the author's analysis is sufficient, but it is not novel or offers new insights.
2. The contrast with adapter method in Section 4.5 is not sufficient. In the pervious work [4], the combination of length adapter and modal adapter is usually used. It seems that the authors only use the modality adapter in therir experiments.
3. The author mentioned that the p-tuning is a parallel method, while the adapter is serial. However, some works [5] also use the parallel adapter structure, showing better results.

[1] “Listen, Understand and Translate”: Triple Supervision Decouples End-to-end Speech-to-text Translation.

[2] Stacked Acoustic-and-Textual Encoding Integrating the Pre-trained Models into Speech Translation Encoders.

[3] Improving Speech Translation by Cross-Modal Multi-Grained Contrastive Learning.

[4] UPC’s Speech Translation System for IWSLT 2021, arXiv:2105.04512v1

[5] Pretrained Speech Encoders and Efficient Fine-tuning Methods for Speech Translation: UPC at IWSLT 2022. https://aclanthology.org/2022.iwslt-1.23.pdf

**Reproducibility:**

4: Could mostly reproduce the results, but there may be some variation because of sample variance or minor variations in their interpretation of the protocol or method.

**Reviewer Confidence:**

5: Positive that my evaluation is correct. I read the paper very carefully and I am very familiar with related work.

---

> ### Author Rebuttal · Authors · 2023-08-29
>
> Thank you for the review.
>
> > Q4.1 *There have been previous analyses of the network representation problem [1,2,3]. I also admit that the author's analysis is sufficient, but it is not novel or offers new insights.*
>
> Thank you for bringing this to our attention. While we recognize and respect the contributions made by the cited works, we believe our research offers a distinct perspective in its approach and findings.
>
> - *Preliminary Analysis Before Method Validation*: Unlike [1] and [2], which introduce methods prior to their analyses, our study places a significant emphasis on initiating with an in-depth preliminary analysis. This approach allows us to deeply understand the underpinnings of the problem, which subsequently guides our method validation.
>
> - *In-Depth Exploration of the Encoder*: The work by [3] underscores the complexity and significance of the ST encoder. However, our analysis delves deeper, exploring both the acoustic and linguistic intricacies of the encoder. This granular assessment not only reaffirms the encoder's importance but also unravels nuances previously unexplored.
>
> - *Novel Insights for the Community*: The unique vantage point from which we approached our research has enabled us to unearth fresh insights. These findings have the potential to catalyze further advancements in speech translation and enrich the broader discourse on speech-text bimodality.
>
> **References**
> [1] Dong et al., Listen, understand and translate: Triple supervision decouples end-to-end speech-to-text translation. AAAI2021.
> [2] Zhang et al., Improving Speech Translation by Cross-Modal Multi-Grained Contrastive Learning. TASLP2023.
> [3] Xu et al., Stacked acoustic-and-textual encoding: Integrating the pre-trained models into speech translation encoders. ACL2021.
>
> > Q4.2 *The contrast with adapter method in Section 4.5 is not sufficient. In the previous work [1], the combination of length adapter and modal adapter is usually used. It seems that the authors only use the modality adapter in their experiments.*
>
> While we appreciate the potential advantages of a length adapter, incorporating it currently falls outside the scope of our research. In Section 4.5, we focus on confirming our observation that higher encoder layers are less effective at processing linguistic features. To maintain a fair comparison, we align our methodology with a default model configuration, which doesn't include a length adapter. Instead, it employs an encoder-to-decoder projection to address dimension discrepancies between the encoder and decoder.
>
> Through our research, we strategically positioned series adapters within specific tiers of the encoder layer. Empirical results showcase the efficacy of this approach, particularly when adapters are located at top layers. This strategy underlines our observations, promotes a clearer understanding, and facilitates a meaningful comparison with other methods.
>
> We hope this clarifies our rationale and the scope of our study.
>
> **References**
> [1]  Gállego et al., Upc’s speech translation system for iwslt 2021. IWSLT2021
>
> >Q4.3 *The author mentioned that the p-tuning is a parallel method, while the adapter is serial. However, some works [1] also use the parallel adapter structure, showing better results. Since the parallel adapter also shows good results, I wonder how the author proposed approach differs in this case.*
>
> Thank you for pointing out the contributions of parallel adapter structures as seen in [1]. In our study, the distinction between serial and parallel methods isn't intended as a hierarchy of effectiveness but rather as two distinct methodologies to validate our observations.
>
> While we acknowledge the demonstrated efficacy of the parallel adapter structure in [1], our primary focus in this paper was to employ both parallel (p-tuning) and serial (modal adapter) methods to substantiate specific findings about the behavior and characteristics of the encoder layers. The intention was not necessarily to claim one approach as superior to the other but to shed light on how each method complements our understanding.
>
> That said, we truly value your suggestion on the potential synergy between serial and parallel methods. This presents an exciting avenue for future research, and we are keen on exploring how a joint approach might further enhance speech translation outcomes.
>
> **References**
> [1]Tsiamas et al., Pretrained Speech Encoders and Efficient Fine-tuning Methods for Speech Translation: UPC at IWSLT 2022.  IWSLT2022

---

### Official Review · Reviewer_A846 · 2023-08-11

**Soundness:** 2

**Excitement:**

3: Ambivalent: It has merits (e.g., it reports state-of-the-art results, the idea is nice), but there are key weaknesses (e.g., it describes incremental work), and it can significantly benefit from another round of revision. However, I won't object to accepting it if my co-reviewers champion it.

**Paper Topic And Main Contributions:**

This paper focuses on the analysis and improvement on end-to-end S2T models when directly finetuned with pretrained speech encoder and text translation decoder, to mitigate the modality gap. The authors first conduct an analysis on several linguistic tasks and find the differences between S2T and T2T models. Then they propose a new model called PromptST to improve the performance following  P-Tuning V2. The results on CoVoST-v2 datasets show the consistent improvement over baselines in terms of both S2T performance and the linguistic evaluation benchmark.

The contributions include:
1. Introducing a linguistic probing benchmark, nameed Speech-Senteval, for speech models.
2. An interesting analysis on comparisons between S2T and T2T models for linguistic tasks.
3. Proposing PromptST for S2T with improvement over baselines.

**Questions For The Authors:**

1. There is another method to initialize the S2T models with pretrained speech model and text translation model: leveraging speech encoder with text encoder on top of it for S2T encoder initialization, and text decoder for S2T decoder initialization (Xu et al. 2021; Fang et al. 2022; ...). Have you tested the results of this method?

References:

Xu et al. 2021: Stacked Acoustic-and-Textual Encoding: Integrating the Pre-trained Models into Speech Translation Encoders, ACL2021

Fang et al. 2022: STEMM: Self-learning with Speech-text Manifold Mixup for Speech Translation, ACL2022

**Reasons To Accept:**

1. Introducing a linguistic probing benchmark, nameed Speech-Senteval, for speech models  and conducting an interesting analysis on comparisons between S2T and T2T models accordingly.

**Reasons To Reject:**

1. The proposed method seems not aligned with the analyzed resulsts on linguistic evaluation. How can the prompt representations improve the linguistic information learning on top encoder layers? It lacks the corresponding analysis between their proposed model PromptST and the baseline, like similar results in Figure 1. If we only see the results of Table 5, the improvement is not consistent and we cannot see the results with the middle layers. Although the final translation results are improved, it is not clear how the improvement come. Is it only from the extra parameters brought by the prompt embeddings?

2. The method is only evaluated on CoVoST-v2. Another polular dataset, MUST-C, is not evaluated, which makes the results less persuasive.


**Reproducibility:**

4: Could mostly reproduce the results, but there may be some variation because of sample variance or minor variations in their interpretation of the protocol or method.

**Reviewer Confidence:**

4: Quite sure. I tried to check the important points carefully. It's unlikely, though conceivable, that I missed something that should affect my ratings.

---

> ### Author Rebuttal · Authors · 2023-08-29
>
> Thank you for the review.
>
> > Q3.1 *The proposed method seems not aligned with the analyzed resulsts on linguistic evaluation. How can the prompt representations improve the linguistic information learning on top encoder layers? It lacks the corresponding analysis between their proposed model PromptST and the baseline, like similar results in Figure 1. If we only see the results of Table 5, the improvement is not consistent and we cannot see the results with the middle layers. Although the final translation results are improved, it is not clear how the improvement come. Is it only from the extra parameters brought by the prompt embeddings?*
>
> Based on our analysis, we suspect that higher-level layers in the model may struggle to capture linguistic nuances, possibly due to limitations in their capacity. Previous research [1-3] supports the idea that enhanced representations can improve translation quality. In light of this, we introduced additional features to the upper layers of the encoder to augment their representational power.
>
> For our proposed PromptST method, we did conduct a detailed analysis, similar to what is presented in Figure 1. Due to space constraints in the current paper, these specific findings are not included. However, we are considering incorporating them in the next version for a more complete understanding. The table below presents the results based on models using the PromptST approach.
> *****************************************************************
> |            |                        |     Emb    |        4       |      8     |       12       |       16       |     20     |       24       |
> |:----------:|:----------------------:|:----------:|:--------------:|:----------:|:--------------:|:--------------:|:----------:|:--------------:|
> |   Surface  |      Word Content      |     0.4    |       1.1      |     2.5    |      14.4      |      11.8      |    14.2    |       4.4      |
> |            |     Sentence Length    |    76.5    |      68.1      |    76.5    |      77.3      |      79.5      |    80.5    |      78.9      |
> | Synatactic |     Top Constituent    |    32.3    |      39.3      |    48.4    |      62.2      |      67.6      |    61.9    |      54.3      |
> |            |       Tree Depth       |    19.7    |      28.5      |    28.0    |      36.0      |      31.7      |    31.2    |      39.5      |
> |            |      Bigram Shift      |    45.5    |      49.7      |    57.2    |      58.7      |      71.2      |    67.3    |      69.6      |
> |  Semantic  | Coordination Inversion |    54.7    |      46.3      |    51.2    |      61.5      |      59.8      |    57.3    |      61.6      |
> |            |      Object Number     |    65.2    |      78.9      |    80.4    |      78.1      |      81.2      |    78.1    |      82.5      |
> |            |      Past Present      |    67.2    |      69.0      |    78.0    |      81.6      |      80.8      |    83.8    |      81.9      |
> |            |     Subject Number     |    53.7    |      65.6      |    70.4    |      79.9      |      78.8      |    81.8    |      83.8      |
> |            |       Odd Man Out      |    55.6    |      50.9      |    47.1    |      49.8      |      52.9      |    53.9    |      55.6      |
> **********************************************
> Our experimental data clearly indicate that using PromptST enhances the model's ability to retain complex linguistic information. We specifically observe improvements in capturing surface and syntactic properties at layers 12, 16, and 24, with increases of +3.2, +1.9, and +1.4, respectively. These findings suggest that PromptST effectively expands the learning capacity of the model's higher layers, positively influencing the types of linguistic properties that the encoder can learn.
>
> **References:**
> [1] Wu et al., Pay less attention with lightweight and dynamic convolutions. ICLR2019.
> [2] Wei et al., Multiscale collaborative deep models for neural machine translation.  ACL2020.
> [3] Wang et al., Learning deep transformer models for machine translation. ACL2019.
>
> > Q3.2 *The method is only evaluated on CoVoST-v2. Another polular dataset, MUST-C, is not evaluated, which makes the results less persuasive.*
>
> Following your constructive suggestion, we have conducted a preliminary evaluation of PromptST on the MuST-C en-de tst-COMMON dataset, and the results are tabulated as follows:
> *******************************************
> | Continue Train | PromptST |
> |:--------------:|:--------:|
> |      16.8      | **17.1** |
> **************************************
> It's important to note that, given our time limitations, this evaluation utilizes a model trained on CoVoST training data, not the MuST-C training data. Despite this constraint, the initial findings are promising, showcasing that PromptST achieves a marginally higher performance than the continue-trained model. We plan to undertake more comprehensive experiments in the future to substantiate our observations.
>
> > Q3.3 *There is another method to initialize the S2T models with pretrained speech model and text translation model. Have you tested the results of this method?*
>
> Thank you for bringing this to our attention, which we had previously overlooked. We've conducted further experiments to test our method when initialized with other pre-trained models, but these are still in progress due to time constraints. Nonetheless, we remain confident that our approach is applicable to various pre-trained models.
>
> Our research focuses on a model that is already well-regarded and widely used in the speech-to-text field. We begin by situating it within a broader framework, followed by an in-depth performance analysis. We believe that the contributions of our research go beyond the specific method we've developed. This includes the new benchmark we introduce and the broader conclusions we draw, all of which are designed to make meaningful contributions to future research in this domain.

---

### Official Review · Reviewer_gne1 · 2023-08-11

**Soundness:** 4

**Excitement:**

3: Ambivalent: It has merits (e.g., it reports state-of-the-art results, the idea is nice), but there are key weaknesses (e.g., it describes incremental work), and it can significantly benefit from another round of revision. However, I won't object to accepting it if my co-reviewers champion it.

**Paper Topic And Main Contributions:**

This work first proposes to analyze S2T (speech-to-text) encoder by probing acoustic and linguistic information. A comparison with T2T shows that upper encoder layer might not learn linguistic information efficiently. Motivated by this observation, they proposes to attach soft prompt to the upper encoder layers to be jointly fine-tuned with the decoder. The encoder of proposed model achieves a better performance on the linguistic probing tasks.

**Reasons To Accept:**

- This work is well-motivated by its observation that upper layer of encoder does not encode linguistic information efficiently. The proposed soft prompt is able to address this problem to some extent.
- it contains several detailed analysis which helps more to understand the model. personally I find the analysis of prompt length to be very interesting
- it proposes a new dataset/benchmark to probe information from S2T model

**Reasons To Reject:**

- while I understand the motivation of the proposed approach, I am not sure what's the benefit of only using efficient approach on the encoder side. The model seems to fix the encoder but fully fine-tune the decoder if I understand correctly. One of the main reasons of prompt learning is to significantly reduce the trainable parameters during the fine-tuning, but this benefit is not very clear if decoder can be fully fine-tuned. (still half of the parameters are trainable). why not just fully fine-tune the entire model instead?
- The baseline models are not very strong as the proposed model simply contains more tunable parameters compared with model 5,6. It is expected to perform better than them by nature. To more highlight the benefit of prompt,  It might be better to have experiments such as 0) prompt both encoder and decoder but fixing encoder/decoder 1) fine-tune prompt only by fixing both encoder and decoder. 2) fine-tune all prompt, encoder, decoder 3) fine-tune encoder/decoder without prompt

**Reproducibility:**

4: Could mostly reproduce the results, but there may be some variation because of sample variance or minor variations in their interpretation of the protocol or method.

**Reviewer Confidence:**

3: Pretty sure, but there's a chance I missed something. Although I have a good feel for this area in general, I did not carefully check the paper's details, e.g., the math, experimental design, or novelty.

---

> ### Author Rebuttal · Authors · 2023-08-29
>
> Thank you for the review.
>
> > Q2.1 *I am not sure what's the benefit of only using efficient approach on the encoder side. The model seems to fix the encoder but fully fine-tune the decoder if I understand correctly. One of the main reasons of prompt learning is to significantly reduce the trainable parameters during the fine-tuning, but this benefit is not very clear if decoder can be fully fine-tuned. why not just fully fine-tune the entire model instead?*
>
> Thank you for raising this insightful question regarding our model fine-tuning strategy. In our study, as detailed in Section 3, we underscore the pivotal role of the encoder, particularly given its responsibility for managing intricate tasks. From our analyses, we've observed that the higher layers of pre-trained encoders might not be optimally primed to encapsulate linguistic intricacies inherent to speech translation (ST).
>
> Our experimental journey provided several insights:
> 1. *Whole Model Fine-tuning*: Our initial attempts at fine-tuning the entire model illuminated two major issues. First, it proved to be considerably resource-intensive. Second, and more crucially, the results were not as promising as anticipated.
> 2. *Prompt-only Fine-tuning*: Given the findings from our initial experiment, we explored fine-tuning only the prompt. However, this approach didn't yield satisfactory performance, potentially stemming from the multifaceted nature of the ST challenge. It became apparent that solely adjusting the prompt couldn't unlock the model's full potential.
> 3. *Focused Fine-tuning Strategy*: Based on our earlier learnings, we formulated an optimized strategy. We decided to hold the encoder constant, but fully fine-tune both the decoder and the prompt. To provide a perspective on parameter efficiency: the combined parameters of the decoder and the prompt amount to 17 million, a mere fraction when compared to the encoder's substantial 310 million parameters.
> This fine-tuned strategy, while being more resource-conservative, also culminated in markedly improved fine-tuning outcomes. Thus, our approach harmoniously melds efficiency with performance, paving the way for successful ST model deployment.
>
> **References**
> [1] Xu et al. Stacked Acoustic-and-Textual Encoding: Integrating the Pre-trained Models into Speech Translation Encoders. ACL2021.
>
>
> > Q2.2 *The baseline models are not very strong as the proposed model simply contains more tunable parameters compared with model 5,6. It is expected to perform better than them by nature. To more highlight the benefit of prompt, It might be better to have experiments such as 0) prompt both encoder and decoder but fixing encoder/decoder 1) fine-tune prompt only by fixing both encoder and decoder. 2) fine-tune all prompt, encoder, decoder 3) fine-tune encoder/decoder without prompt.*
>
> Thank you for the thoughtful feedback, and we concur with your viewpoint that assessing our method against a variety of tuning strategies is essential for a holistic understanding.
>
> To address your concerns, we embarked on comprehensive experiments for the English-to-German translation task. Here's a breakdown of our findings:
>
> ***********************************************************
> |             **Strategy**            |   BLEU |
> |:------------------:|:--------:|
> |     finetune encoder and prompt     |   23.6   |
> |           finetune encoder          |   23.8   |
> |     finetune encoder and decoder    |   25.5   |
> |           finetune decoder          |   25.9   |
> |           finetune prompt           |   25.9   |
> | finetune encoder decoder and prompt |   26.0   |
> |     finetune decoder and prompt     | **26.4** |
> ************************************************************
>
> These experiments solidify our understanding in several ways:
> - *Isolated Encoder Fine-tuning*: When solely fine-tuning the encoder, we observed a tendency for rapid overfitting, indicating that this alone might not be the best strategy.
> - *Synergistic Approach*: Our original method of fine-tuning the decoder and prompt showcased the best performance. This aligns with our hypothesis that integrating prompts with the decoder's tuning can provide a balance, enhancing model generalization while mitigating overfitting tendencies.
> - *Prompt's Efficacy*: When tuning only the prompt, the model delivered competitive results, underscoring the prompt's potential to improve performance even without adjusting the core model components.

---

### Official Review · Reviewer_aEu5 · 2023-08-11

**Soundness:** 4

**Excitement:**

4: Strong: This paper deepens the understanding of some phenomenon or lowers the barriers to an existing research direction.

**Missing References:**

- As a minor suggestion, additional references to related findings in previous work would improve the paper, for example, the observation that the modality gap is noticeable in top decoder layers which are important to translation quality (Tang et al. 2021 "Improving Speech Translation by Understanding and Learning from the Auxiliary Text Translation Task"), or the hierarchy of linguistic information in Transformer encoder's hidden representation (Lin et al. 2019 "Open Sesame: Getting inside BERT’s Linguistic Knowledge", Rogers et al. 2020 "A Primer in BERTology: What We Know About How BERT Works").
- L473: Le et al. 2021 (Lightweight Adapter Tuning for Multilingual Speech Translation) investigates the use of both sequential and parallel adapters for speech translation.

**Paper Topic And Main Contributions:**

This paper investigates the acoustic and linguistic behaviors of ST models using a linguistic probing benchmark newly proposed in this paper. By understanding how the acoustic encoder encodes different types of acoustic and linguistic knowledge, the paper proposes a plug-in prompt-learning strategy to better fuse the speech and text modalities in ST model.

**Questions For The Authors:**

- Why is the choice of 7 layers for the decoder?
- Why continued training gives much better result for En-Ca compared to remaining pairs?
- How the performance of acoustic features are evaluated in Figure 2?
- It would be interesting to evaluate the performance of this prompting-based plug-in in another popular ST benchmark such as MuST-C where many strong ST systems report the results on.
- Do you think this approach is complementary to existing strong approaches in building ST systems such as multi-task learning?

**Reasons To Accept:**

- The paper offers an interesting analysis that helps strengthen understandings of the behaviors of acoustic encoder in the ST model.
- The release of a new probing benchmark would facilitate the analysis of speech encoding methods in the future.
- The use of prompt learning for ST is novel and well-justified.

**Reasons To Reject:**

1) The literature review of methods to mitigate the speech-text modality gap or speech-text fusion techniques seems to be too brief (L57-59). For example, methods such as the increasingly common cascaded encoders approach (Liu et al. 2020 "Bridging the Modality Gap for Speech-to-Text Translation", Xu et al. 2021 "Stacked Acoustic-and-Textual Encoding: Integrating the Pre-trained Models into Speech Translation Encoders"), using implicit guidance from external ASR model (Yin et al. 2023 "Improving speech translation by fusing speech and text"), aligning speech and text sequences at the supervised pre-training stage (Le et al. 2023 "Pre-training for Speech Translation: CTC Meets Optimal Transport"), or joint speech-text pre-training approaches (Bapna et al. 2021 "SLAM: A Unified Encoder for Speech and Language Modeling via Speech-Text Joint Pre-Training", Tang et al. 2022 "Unified Speech-Text Pre-training for Speech Translation and Recognition"), etc.
2) Limited comparison to parameter-efficient techniques other than adapter-tuning, for example LNA tuning (Li et al., 2020 "Multilingual speech translation with efficient fine-tuning of pre-trained models"), MAM adapter (He et al. 2022 "Towards a Unified View of Parameter-Efficient Transfer Learning ").

**Reproducibility:**

3: Could reproduce the results with some difficulty. The settings of parameters are underspecified or subjectively determined; the training/evaluation data are not widely available.

**Reviewer Confidence:**

4: Quite sure. I tried to check the important points carefully. It's unlikely, though conceivable, that I missed something that should affect my ratings.

**Typos Grammar Style And Presentation Improvements:**

- L193: lack of period.
- L224: incomplete sentence
- L243: incorrect period
- L322: "an raw waveform input" -> "a raw waveform input"
- Figure 3: "For a $M$-layers encoder" -> "For an $M$-layers encoder"
- L525: "... by averaging +0.61 BLEU": a bit confusing

---

> ### Author Rebuttal · Authors · 2023-08-29
>
> Thank you for the review.
>
> > Q1.1 *The literature review of methods to mitigate the speech-text modality gap or speech-text fusion techniques seems to be too brief (L57-59). For example, methods such as the increasingly common cascaded encoders approach[1-2], using implicit guidance from external ASR model[3], aligning speech and text sequences at the supervised pre-training stage[4], or joint speech-text pre-training approaches [5-6].*
>
> We appreciate the feedback regarding the brevity of our literature review on methods that address the speech-text modality gap and fusion techniques. Recognizing the importance of these references, we will ensure they are thoroughly discussed in the Introduction and Related Work sections of our revised manuscript.
>
> It's worth emphasizing the unique trajectory our study has taken in contrast to the aforementioned works. Unlike the techniques presented in references [1-5], which prioritize introducing methods prior to embarking on analysis, our methodological approach revolves around a deep analysis that precedes validation of the method. This distinction underscores our key contribution: revealing novel insights into the intricacies of end-to-end models.
>
> In particular, while reference [6] elucidates the complexity and significance of the ST encoder, our research ventures further. We delve into a nuanced exploration of the ST encoder by assimilating both acoustic and linguistic analyses. This depth of investigation, we believe, sets our work apart and adds a unique dimension to the existing body of literature.
>
> **References**
> [1] Liu et al., Bridging the Modality Gap for Speech-to-Text Translation. arXiv2020.
> [2] Yin et al., Improving speech translation by fusing speech and text. arXiv2023.
> [3] Le et al., Pre-training for Speech Translation: CTC Meets Optimal Transport. arXiv2023.
> [4] Bapna et al. SLAM: A Unified Encoder for Speech and Language Modeling via Speech-Text Joint Pre-Training. arXiv2021.
> [5] Tang et al. Unified Speech-Text Pre-training for Speech Translation and Recognition. ACL2022.
> [6] Xu et al. Stacked acoustic-and-textual encoding: Integrating the pre-trained models into speech translation encoders. ACL2021.
>
> > Q1.2 *Limited comparison to parameter-efficient techniques other than adapter-tuning.*
>
> Our primary objective in this study centers on a comprehensive evaluation of the model through diverse probing tasks. Consequently, our emphasis wasn't on drawing comparisons among various fine-tuning techniques. Rather, we anchored our approach on simple yet efficient fine-tuning mechanisms to corroborate our findings.
>
> However, we recognize the potential value of such broader comparisons. Given the robustness of our observations supported by our experiments, we are confident that these insights can be extrapolated and hold promise when integrated with other parameter-efficient techniques. In future iterations of this work, we will certainly consider expanding the comparative spectrum to encompass additional fine-tuning methodologies.
>
> > Q1.3 *Why is the choice of 7 layers for the decoder?*
>
>  We grounded our selection in the wav2vec 2.0+self-training (Wang et al., 2021) baseline, a cornerstone in current speech processing research. By aligning with this recognized model configuration, we aimed to facilitate not only a direct and reliable comparison but also to anchor our work within the best practices of contemporary research.
>
> **References**
> [1] Wang et al., Large-scale self-and semi-supervised learning for speech translation. Interspeech2021.
>
> > Q1.4 *Why continued training gives much better result for En-Ca compared to remaining pairs?*
>
> Our hypothesis for the significant improvement observed for En-Ca, when subjected to continued training, stems from a potential under-convergence in the initial checkpoint provided by Wang et al.(2021) specifically for this language pair. The rapid rate of convergence we witnessed during our extended training phase reinforces this belief. This suggests that the model had more room to optimize for En-Ca, and by granting it additional training time, we were able to harness this potential.
>
> > Q1.5 *How the performance of acoustic features are evaluated in Figure 2?*
>
> To assess the performance of acoustic features as presented in Figure 2, we undergo a multi-step evaluation:
> 1. We begin by extracting frame-level features from every layer of the encoder.
> 2. These extracted features are then employed in a phoneme recognition task, leveraging the TIMIT dataset.
> 3. Subsequently, we train a supervised classifier using the aforementioned features.
> 4. Finally, the trained classifier's performance is gauged on the test dataset.
>
> Recognizing the importance of transparency in methodology, we are committed to providing a more granular and comprehensive description of this process in our revised manuscript.
>
>
> > Q1.6 *Evaluate the performance of this prompting-based plug-in in MuST-C dataset.*
>
> Thank you for your constructive suggestion. Currently, we have conducted a preliminary evaluation of PromptST on the MuST-C en-de tst-COMMON dataset, and the results are tabulated as follows:
>
>
> | Continue Train | PromptST |
> |:--------------:|:--------:|
> |      16.8      | **17.1** |
>
>
> It's important to note that, given our time limitations, this evaluation utilizes a model trained on CoVoST training data, not the MuST-C training data. Despite this constraint, the initial findings are promising, showcasing that PromptST achieves a marginally higher performance than the continue-trained model. We plan to undertake more comprehensive experiments in the future to substantiate our observations.
>
> > Q1.7 *Do you think this approach is complementary to existing strong approaches in building ST systems such as multi-task learning?*
>
> Absolutely. We believe that our approach can be seen as a complementary strategy to the strong methodologies currently being developed for ST models.
>
> Given the pervasive challenge of data scarcity in the domain of speech translation, it's increasingly common to bootstrap models with pre-trained parameters. Notably, the multi-task learning methods as introduced by Ye et al. (2022) and Tang et al. (2021) also employ wav2vec2 for initializing their models, echoing our approach.
>
> What bolsters our confidence in our approach's general applicability is its inherent flexibility. Our method is not only amenable to integration with newer models but also the insights we've derived have the potential to propel further research in this domain. We envision our findings enriching the current repertoire of techniques, potentially leading to breakthroughs in ST system performance.
>
> **References**
> [1] Ye et al., Cross-modal Contrastive Learning for Speech Translation. NAACL2022.
> [2] Tang et al., Improving Speech Translation by Understanding and Learning from the Auxiliary Text Translation Task. ACL2021.

---

### Meta-Review · Area_Chair_xGEX · 2023-08-31

**Recommendation:** 5
**Confidence:** 4

**Metareview:**

**Summary:**
The paper focuses on analyzing and improving end-to-end (E2E) Speech-to-Text (S2T) models by probing acoustic and linguistic information. The authors introduce a linguistic probing benchmark, Speech-Senteval, and propose the PromptST model, which utilizes prompt learning to address modality gap issues. The analysis of linguistic tasks highlights differences between S2T and Text-to-Text (T2T) models. The proposed model improves S2T performance and linguistic evaluation benchmark on CoVoST-v2 datasets.

**Pros:**

- The paper introduces a novel linguistic probing benchmark, Speech-Senteval, to analyse the speech models (Reviewer 1, Reviewer 3).

- The analysis of linguistic tasks between S2T and T2T models contributes to understanding the behaviors of acoustic encoders (Reviewer 1, Reviewer 2).

- The PromptST model's introduction provides an innovative solution for mitigating the modality gap in S2T models (Reviewer 1, Reviewer 2, Reviewer 3, Reviewer 4).

- The experimental validation of the proposed approach on CoVoST-v2 datasets demonstrates consistent performance improvement over baselines (Reviewer 3, Reviewer 4).

- The paper's methodology of analyzing information processing within models and probing the balance between linguistic and acoustic information is insightful (Reviewer 4).

**Cons:**

- The literature review related to methods addressing the modality gap or fusion techniques is deemed rather brief by some reviewers, potentially affecting the paper's novelty and thoroughness (Reviewer 1, Reviewer 2).

- The benefits of using the proposed efficient approach on the encoder side, while fully fine-tuning the decoder, lack clarity and comparison against alternative strategies (Reviewer 2).

- The analysis primarily focuses on linguistic performance, potentially overlooking the broader implications of the proposed model's improvements across various fields (Reviewer 4).

- The absence of an evaluation on the MUST-C dataset limits the persuasiveness and generalization of the results (Reviewer 3).

- The mechanism by which prompt representations enhance linguistic information learning on the upper encoder layers needs further analysis and clarification (Reviewer 3).

Reviewer 1: aEu5,
Reviewer 2: gne1,
Reviewer 3: A846,
Reviewer 4: LoxQ

---

### Decision · Program_Chairs · 2023-10-07

**Decision:**

Accept-Main

**Comment:**

**Summary:**
The paper focuses on analyzing and improving end-to-end (E2E) Speech-to-Text (S2T) models by probing acoustic and linguistic information. The authors introduce a linguistic probing benchmark, Speech-Senteval, and propose the PromptST model, which utilizes prompt learning to address modality gap issues. The analysis of linguistic tasks highlights differences between S2T and Text-to-Text (T2T) models. The proposed model improves S2T performance and linguistic evaluation benchmark on CoVoST-v2 datasets.

**Pros:**

- The paper introduces a novel linguistic probing benchmark, Speech-Senteval, to analyse the speech models (Reviewer 1, Reviewer 3).

- The analysis of linguistic tasks between S2T and T2T models contributes to understanding the behaviors of acoustic encoders (Reviewer 1, Reviewer 2).

- The PromptST model's introduction provides an innovative solution for mitigating the modality gap in S2T models (Reviewer 1, Reviewer 2, Reviewer 3, Reviewer 4).

- The experimental validation of the proposed approach on CoVoST-v2 datasets demonstrates consistent performance improvement over baselines (Reviewer 3, Reviewer 4).

- The paper's methodology of analyzing information processing within models and probing the balance between linguistic and acoustic information is insightful (Reviewer 4).

**Cons:**

- The literature review related to methods addressing the modality gap or fusion techniques is deemed rather brief by some reviewers, potentially affecting the paper's novelty and thoroughness (Reviewer 1, Reviewer 2).

- The benefits of using the proposed efficient approach on the encoder side, while fully fine-tuning the decoder, lack clarity and comparison against alternative strategies (Reviewer 2).

- The analysis primarily focuses on linguistic performance, potentially overlooking the broader implications of the proposed model's improvements across various fields (Reviewer 4).

- The absence of an evaluation on the MUST-C dataset limits the persuasiveness and generalization of the results (Reviewer 3).

- The mechanism by which prompt representations enhance linguistic information learning on the upper encoder layers needs further analysis and clarification (Reviewer 3).

Reviewer 1: aEu5,
Reviewer 2: gne1,
Reviewer 3: A846,
Reviewer 4: LoxQ